# The Global, Regional and National Burden of Pancreatic Cancer Attributable to Smoking, 1990 to 2019: A Systematic Analysis from the Global Burden of Disease Study 2019

**DOI:** 10.3390/ijerph20021552

**Published:** 2023-01-14

**Authors:** Wenkai Jiang, Caifei Xiang, Yan Du, Xin Li, Wence Zhou

**Affiliations:** 1The Second Clinical Medical College, Lanzhou University, No. 222 Tianshui South Road, Cheng-Guan District, Lanzhou 730030, China; 2The First Clinical Medical College, Lanzhou University, Lanzhou 730030, China; 3Department of General Surgery, Lanzhou University Second Hospital, Lanzhou 730030, China

**Keywords:** pancreatic cancer, risk factor, smoking, tobacco, death, disability-adjusted life-year

## Abstract

Objective: Pancreatic cancer poses a serious medical problem worldwide. Studies have reported the relationship between smoking and cancer. This study aimed to evaluate the burden of pancreatic cancer attributable to smoking and its global, regional and national trends, patterns and alterations from 1990 to 2019. Methods: Data were extracted from the Global Health Data Exchange query tool, including deaths, disability-adjusted life-years (DALYs) and age-standardized rates (ASRs). Measures were stratified by sex, age, region, country/territory and sociodemographic index (SDI). We used Joinpoint regression to determine the secular trend of ASRs by calculating the average annual percentage change (AAPC). Results: In 2019, smoking risk-related deaths and DALYs accounted for 21.3% and 21.1% of global pancreatic cancer, respectively. There were 113,384 (95% UI 98,830 to 128,466) deaths of smoking-attributable pancreatic cancer worldwide in 2019, of which 64.1% were in males. The disease burden was higher in males than in females. High-income regions or large population regions had the higher disease burden. East Asia carried the highest number of smoking-attributable pancreatic cancer deaths and DALYs. The Caribbean had the fastest increasing rate (AAPC = 3.849, 95% CI 3.310 to 4.391) of age-standardized death rate over the past 30 years. In 2019, China had the highest number of deaths, which was followed by the USA and Japan. There was a trend of increasing ASDR along with increases in SDI. Conclusion: Variations existed in the smoking risk-related pancreatic cancer burden among different sexes, age groups, regions and countries/territories. The burden of smoking-attributable pancreatic cancer should be considered an important health issue. Future strategies should include comprehensive policies to control tobacco use.

## 1. Introduction

Pancreatic cancer poses a major medical problem worldwide. The past 20 years have seen a doubling in the annual number of pancreatic cancers diagnosed worldwide [1]. The cancer risk factors can be grouped as nonmodifiable and modifiable risk factors. The modifiable risk factors for pancreatic cancer include behavioral factors and metabolic factors. Cigarette smoking is a well-established behavioral risk factor for various cancers, such as tracheal, bronchus and lung cancer, laryngeal cancer, head and neck squamous cell carcinomas, esophageal cancer, colorectal cancer and pancreatic cancer [2,3,4,5,6,7]. Smoking was the second leading risk factor for mortality in both sexes in 2015 [8]. The global age-standardized summary exposure value (SEV) of tobacco use ranked high among all behavioral factors in 2019 [9]. To improve the survival rate of pancreatic cancer patients, it is important to identify significant modifiable risk factors. However, the distribution of risk factors varies in different regions and countries/territories, with the influence degree of the same risk factor on the population of different regions, ages and sexes being different. Therefore, analyzing the disease burden of pancreatic cancer attributable to smoking at different levels can contribute to cancer prevention and public health.

The Global Burden of Disease (GBD) Study 2019 included 369 diseases, injuries and their 87 related risk factors in all regions and countries/territories [9,10]. In this article, we conducted a systematic analysis based on GBD 2019 to explore the epidemiological trends, patterns and alterations of smoking-attributable pancreatic cancer death and disability-adjusted life-year (DALY) features, aiming to better understand regional disparities and provide new insight into cancer prevention and public health.

## 2. Methods

### 2.1. Data Sources

Data on smoking-attributable pancreatic cancer burden were extracted from the Global Health Data Exchange (https://ghdx.healthdata.org (accessed on 31 October 2022)), including deaths, DALYs, age-standardized rates (ASRs), summary exposure values (SEVs), annualized rates of change (ARCs) and their 95% uncertainty intervals (UIs). The data covered four world regions, five sociodemographic index (SDI) quintiles, 21 GBD regions and all countries/territories of different age groups and sexes. The general methods for the GBD 2019 and the methods for estimations of disease burden have been detailed in previous studies [9,10]. Detailed descriptions of the GBD 2019 Study are described in Appendix A. This analysis was performed in accordance with the Guidelines for Accurate and Transparent Health Estimates Reporting [11]. All the information about ethical standards is available through the GBD official website (http://www.healthdata.org/gbd/2019 (accessed on 31 October 2022)).

### 2.2. Risk Factor and Exposure Definition

The comparative risk assessment (CRA) framework was used to estimate the proportion of death and DALY attributable to the risk factor for pancreatic cancer. The CRA framework, developed by GBD Risk Factor Collaborators, measured various behavioral, environmental, occupational, and metabolic risks. Tobacco use was one of the behavioral risks, including smoking, secondhand smoke and chewing tobacco. The GBD study used spatiotemporal Gaussian process regression to model the prevalence of both current and former smoking tobacco use, which are available in the appendix of a previous publication [12]. The prevalence of smoking includes current smoking and former smoking. Cross-sectional nationally representative household surveys were the primary source of exposure data. Smoking cases were defined as individuals who were former and currently using any smoked tobacco product on a daily or occasional basis. The relative risks were estimated from cohort and case-control studies. A Bayesian meta-regression model was used to produce nonlinear dose–response curves [13]. The Preferred Reporting Items for Systematic reviews and Meta-Analyses diagrams for each outcome can be found in the previous publication [12]. Data inputs for exposure for smoking included 3439 sources from 201 countries, and relative risks for smoking included 673 sources from 16 countries [9]. The smoking relative risk of pancreatic cancer is presented in Appendix A.

### 2.3. SDI

The SDI is a comprehensive indicator of the development status of a country or region. It is based on the overall fertility rate among women under 25 years old, the mean education level of individuals aged 15 and older, and the lag-distributed income per capita [14]. Countries were divided by SDI into five categories (low, low–middle, middle, high–middle, and high). More SDI information and SDI values by location can be found in Appendix A.

### 2.4. Measures Estimation

The SEV ranges from 0% to 100%, where 100% means the entire population is at the maximum level for one risk and 0 means there is no risk exposure in a population [14]. The age-standardized SEVs across the age groups were used to evaluate the risk exposures. The ARCs were used to assess the change trends of SEVs.

The ASR, which was calculated based on the world standard population, is a measure that can eliminate the influence of population age structure differences to the greatest extent [7]. The ASR was calculated based on the following formula:(1)$$ASR=\left(\overset{A}{\underset{i=1}{\sum }}a_{i}w_{i}/\overset{A}{\underset{i=1}{\sum }}w_{i}\right)\times 100,000$$

The annual percentage change (APC) was used to estimate the rate of change in a given time period. The average annual percent change (AAPC), which provides a summary measure of the APCs over a period of time where the trend is not constant, was used to assess the trends in the incidence and mortality data of cancer disease [15,16]. The APC and AAPC were calculated based on the following formulas:(2)$$APC=\left[\frac{y_{x_{1}}-y_{x}}{y_{x}}\right]\times 100=\left(e^{\beta 1}-1\right)\times 100$$
(3)$$AAPC=(exp(\frac{\sum w_{i}\beta_{i}}{\sum w_{i}})-1)\times 100$$
(*y*: rate; *x*: year; *β*_1_: regression coefficient; *w_i_*: the number of years included in the interval; *β_i_*: regression coefficients corresponding to each interval).

### 2.5. Statistical Analysis

All tests and calculations were performed by using R software (Version 4.2.1). Temporal trends were estimated by APCs, AAPCs and their 95% confidence intervals (CIs) using Joinpoint software (Version 4.9.1.0) [16]. Correlations were determined using Pearson correlation tests. We assessed the correlation between AAPCs from 1990 to 2019 and ASRs in 2019, and the human development index in 2019 (HDI, data source: http://hdr.undp.org/en/composite/HDI (accessed on 31 October 2022)), respectively. Associations between ASRs and SDI in all GBD regions and countries/territories were also assessed. Hierarchy cluster analysis was conducted to group all countries/territories into different categories in terms of their temporal trends in ASRs. The “ggplot2” package of R software was used for visualization of all data. All rates are reported per 100,000 person-years. Data are presented as values with their 95% CIs or 95% UIs. A *p*-value of less than 0.05 was considered statistically significant.

## 3. Results

### 3.1. SEV and ARC of Tobacco from 1990 to 2019

The age-standardized SEV rate of tobacco use decreased globally (ARC = −0.25, 95% UI: −0.26 to −0.24), from 14.85 (95% UI: 13.27 to 16.56) in 1990 to 11.14 (95% UI: 9.93 to 12.54) in 2019. Among the five SDI quintiles, the high SDI and high-middle SDI region had the highest age-standardized SEV rate in 1990 and 2019, respectively. The GBD region with the highest age-standardized SEV rate changed from high-income North America (ASR = 23.3, 95% UI: 21.18 to 25.56) to Central Europe (ASR = 17.7, 95% UI: 15.71 to 19.93). The age-standardized SEV rates and ARC of smoking among all regions are shown in Table 1.

### 3.2. Burden at the Global and Four World Region Levels

In 2019, smoking risk-related deaths and DALYs accounted for 21.3% and 21.1% of all pancreatic cancer deaths and DALYs, respectively. There were 113,384 (95% UI 98,830 to 128,466) deaths of smoking-attributable pancreatic cancer globally in 2019, of which 64.1% were in males (72,682, 95% UI 60,495 to 85,113). This proportion was 60.4% in 1990. There was a 2.0-fold increase in DALYs due to smoking-attributable pancreatic cancer, increasing from 1.22 million (95% UI 1.08 to 1.35) in 1990 to 2.44 million (95% UI 2.11 to 2.77) in 2019. The age-standardized death rates (ASDRs) did not show a significant change globally from 1990 to 2019 (AAPC = −0.028, 95% CI −0.136 to 0.079, *p* = 0.606). The age-standardized DALY rate showed a downward trend globally over the past 30 years (AAPC = −0.131, 95% CI −0.226 to −0.035, *p* = 0.007).

The number of deaths peaked at the ages of 65 to 69 years in males, whereas the peak in females was observed at the ages of 70 to 74 years (Appendix A). The numbers of deaths and DALYs were lower in females younger than 85 years than in males in the same age group, whereas the numbers were higher in females than in males in the age group of 85+ years. The proportion of deaths and DALYs attributable to smoking showed downward trends in both sexes, males and females, from 1990 to 2019 (Appendix A). In 2019, the proportion of pancreatic cancer attributable to smoking use varied by age group and sex (Appendix A). For instance, the proportion of deaths attributable to smoking was higher than 25% in males aged between 55 and 79 years; however, smoking accounted for approximately 15% to 20% of pancreatic cancer deaths among females in the same age group.

For the four continents, Asia had the highest deaths in 2019 (48,434, 95% UI 40,472 to 56,857). The ASDRs and age-standardized DALY rate showed upward trends in Africa, Asia and Europe but a downward trend in America from 1990 to 2019 (Figure 1 and Appendix A). Age-specific rates for deaths increased with increasing age both in males and females, but the DALY rates declined after different age nodes. In addition, ASDRs and age-standardized DALY rates are consistently higher in males than in females among all ages in Africa, Asia and Europe. For people in America, after the age of 75 to 79, women have a higher ASDR and age-standardized DALY rate than men (Figure 2).

### 3.3. Burden at the GBD Region Level

In 2019, East Asia carried the highest number of smoking-attributable pancreatic cancer deaths (27,348, 95% UI 21,521 to 33,938) and DALYs (631,986, 95% UI 493,276 to 800,825), which was followed by Western Europe and high-income North America. The highest ASDR region changed from high-income North America in 1990 (3.08, 95% UI 2.62 to 3.54) to Central Europe in 2019 (2.89, 95% UI 2.46 to 3.4) (Table 2). The Caribbean had the fastest increasing rate (AAPC = 3.849, 95% CI 3.310 to 4.391) of ASDR, while the downward trend was most pronounced in Australasia (AAPC = −1.331, 95% CI −1.496 to −1.166) (Table 3). The information about DALYs and age-standardized DALY rates among different GBD regions is shown in Appendix A.

In 2019, the proportion of deaths attributable to smoking varied among different regions. For males, the highest and lowest proportions were observed in Eastern Europe (32.8%, 95% UI 28.2% to 37.1%) and Western Sub-Saharan Africa (8.3%, 95% UI 5.7% to 10.8%) in 2019. For females, the region with the highest proportion was high-income North America (30.9%, 95% UI 23.9% to 38.2%) (Figure 3).

### 3.4. Burden at the Country/Territory Level

China had the highest deaths (26,551, 95% UI 20,856 to 33,174) and DALYs (612,359, 95% UI 474,505 to 779,859) in 2019, which was followed by the USA and Japan (Figure 4). All ASDRs and age-standardized DALY rates varied among different countries/territories in 2019 (Figure 5). The highest ASDR was observed in Greenland (6.68, 95% UI 5.24 to 8.31) in 2019. The age-standardized DALY rates followed a very similar pattern: Greenland was the top in 2019. Ethiopia had the lowest ASDRs and age-standardized DALY rates of smoking-attributable pancreatic cancer in 2019. As shown in Appendix A, the ASDR increased the most in Kazakhstan (AAPC = 6.469, 95% CI 5.793 to 7.149) and decreased the most in Colombia (AAPC = −2.389, 95% CI −3.008 to −1.767). Information on deaths and DALYs among all countries/territories in 1990 and 2019 is shown in Appendix A. All countries/territories with similar AAPC of death and DALY rates in 2019 can be grouped into five clusters (very significant increase, significant increase, minor increase, remained stable or minor decrease and significant decrease), which are shown in Appendix A.

As shown in Appendix A, a significant association was detected between AAPC and ASR (in 1990), and HDI (in 2019), respectively. The ASDR in 1990 reflects the disease reservoir at baseline, and there is a significant negative association between AAPCs and ASDRs (r = −0.522, *p* < 0.001). The HDI in 2019 can represent the level and availability of health care, and a significant negative association can be found between AAPCs and the HDI in each country/territory (r = −0.191, *p* = 0.01).

### 3.5. Burden among Different SDIs

From 1990 to 2019, the SDI quintile with the highest deaths was always high SDI (in 1990: 28,901, 95% UI 25,403 to 32,273; in 2019: 45,959, 95% UI 39,751 to 52,864). Except for the high SDI region, the ASDR increased in other SDI quintiles (AAPC for high SDI = −0.486, 95% CI −0.586 to −0.386). The ASDR and age-standardized DALY rate trends of the five SDI quintiles from 1990 to 2019 are shown in Appendix A.

Figure 6 demonstrates the trend in ASDRs across the SDI by 21 GBD regions from 1990 to 2019 (r = 0.831, *p* < 0.001). In high-income regions, such as high-income North America, although the ASDR continues to decline, it is still much higher than the expected level in all years. In contrast, the ASDRs of the Australasia and high-income Asia Pacific showed downward trends as the SDI increased and finally were either near or lower than the expected level. When the SDI rises to approximately 0.8, the ASR shows a downward trend. The trend in age-standardized DALY rates across the SDI by region from 1990 to 2019 showed a similar pattern (Appendix A).

As shown in Figure 7, the relationship between ASDRs and SDI showed that a country/territory with a higher SDI usually has a higher ASDR of smoking-attributable pancreatic cancer. Similar to the results mentioned previously, the observed levels were much higher than expected in many countries, including Greenland, Monaco and Montenegro. ASDRs in several countries were also much lower than the expected level, such as Singapore, Bahamas and Oman. The relationship between age-standardized DALY rates and SDI among all countries/territories in 2019 has a similar pattern, which is shown in Appendix A.

## 4. Discussion

The global burden of pancreatic cancer has increased rapidly over the past few decades, and pancreatic cancer is expected to be the leading cause of cancer-related mortality [1]. Genetic factors and modifiable exposures play an important role in pancreatic cancer risk [17]. Understanding potentially modifiable risk factors can contribute to prevention efforts, including reducing exposure and identifying individuals most at risk. This approach will help to reduce the growing disease burden of pancreatic cancer. In this study, we selected smoking, one lifestyle risk factor with the strongest correlation with pancreatic cancer, and explored its differences attributed to deaths at regional, national, SDI, age and gender levels. The results showed that the disease burden of smoking-attributable pancreatic cancer remains high; the large differences in smoking-attributable pancreatic cancer deaths and DALYs between sexes, ages, regions, countries/territories and SDIs remind us to investigate genetic factors, risk factors, medical technologies and other underlying biological and sociological issues that contribute to human disease.

Smoking is a recognized risk factor for cancer. A meta-analysis of 12 case-control studies found that current smokers had a 74% increased risk of pancreatic cancer compared with those who had never smoked [18]. Smokeless tobacco has also been linked to the risk of pancreatic cancer [19]. In addition, the number of years smoking was negatively correlated with the survival rate of pancreatic cancer patients [20]. Of the 7,000 chemicals found in tobacco smoke, at least 250 are harmful, and 60 are carcinogenic [5]. Cigarette smoke promotes the metastasis of pancreatic cancer by upregulating the expression of MUC4 in pancreatic cancer tissue [21]. Substances such as polycyclic aromatic hydrocarbons, heterocyclic aromatic amines and metals in tobacco play roles in the progression of pancreatic cancer. In animal models, nicotine promotes pancreatic cancer by inducing dedifferentiation of acinus cells by downregulating GATA6 [22].

Although the impact of smoking on the pancreatic cancer burden varies by region and country/territory, the global burden of smoking-attributable pancreatic cancer is higher in males than in females. This difference should be considered in cancer prevention and early screening. The lower incidence of pancreatic cancer cases and deaths in women worldwide may be related to the fact that women smoke less [23]. In 2015, 82.3% of the world’s daily smokers were men; the age-standardized prevalence of daily smoking is approximately five times higher among men than among women (25.0% vs. 5.4%) worldwide [8]. The tobacco epidemic is concentrated among men, mainly in large population countries in Asia, while the top three countries in terms of female smokers account for only 27.3% of the world’s female smokers [8]. Cumulative smoking exposure was significantly associated with an increased risk of pancreatic cancer in males, while there was no significant association in females [24]. In addition, the antiestrogenic effects of smoking may be another reason for the sex difference [25]. Increased estrogen exposure may reduce the risk of pancreatic cancer in women; women who smoke cigarettes are relatively estrogen deficient, and smoking may alter estradiol metabolism [26]. Tobacco type and dose may also play a role in the sex difference, but more research is needed to confirm this. Additionally, taking into account regional differences, it is necessary for high-income North America and Western Europe females and East Asia and Eastern Europe males to reduce their high tobacco use risk exposure.

The highest global burden of pancreatic cancer attributable to smoking occurs between the ages of 55 and 79. Cohort studies in several countries confirm that smoking prevalence increases with age, peaking in middle age and then declining [27,28]. Smoking-caused cancer is a chronic process as the individual ages and the carcinogens in the body accumulate, usually in the elderly stage, causing cancer and leading to death. No trend in risk was observed for age at starting cigarette smoking, but after smoking for more than 40 years, the risk of pancreatic cancer increased 2.4 times [29]. In addition, it is also necessary to strengthen the control of tobacco use among young people to prevent more people from being exposed to tobacco.

In 2003, the World Health Organization Framework Convention on Tobacco Control was adopted, redefining approaches to tobacco control and policy [30]. Despite continued progress in tobacco control in some countries over the past decade, the global burden of pancreatic cancer deaths attributable to smoking remains high. This may be related to population growth and aging in some countries, such as China and the USA, suggesting that population growth and aging have offset the reduction in the burden of disease in some areas caused by the decline in smoking. This is consistent with the conclusion from GBD 2015 Tobacco Collaborators that unless measures to control and prevent smoking can be significantly accelerated, population-based factors will continue to contribute to the global disease burden of pancreatic cancer [8]. In addition, social, historical and cultural factors in some regions and countries/territories contribute to a higher population exposure to smoking. However, diagnostic screening models and cancer treatments for pancreatic cancer are rapidly developed in high-income regions, so some indicators of the disease burden are declining compared with 1990.

Tobacco exposure remains higher in high SDI and high-middle SDI regions, including high-income Asia Pacific, Europe and East Asia. In particular, the age-standardized SEV was higher than that in 1990 in Central Europe. Over the past 30 years, rapid changes in the social status and economic strength of these regions, including increasing wealth and leisure opportunities, have contributed to tobacco use, resulting in a high disease burden. Moreover, it is well known that smoking results in an acute mood “boost”, including increased positive affect and decreased negative affect [31]. In regions with rapid economic development, more populations are engaged in finance, electronics and software industries. These people have a faster pace of life and greater work pressure, resulting in smoking to relieve pressure and facilitate a soothing mood. Thus, some high-stress industries should introduce other healthy stress-relieving activities into their daily routine.

Differences in pancreatic cancer death and DALY rates attributable to smoking among countries/territories also reflect differences in epidemiological registry data and methods of diagnosis of pancreatic cancer. Since 1990, regional and national ASDRs have generally increased with the increase in SDI. Over the past three decades, the burden of disease has been higher in high SDI areas and lower in low SDI. The higher burden of pancreatic cancer disease in countries with high SDI may be due to population aging and lifestyle choices that increase exposure to risk factors; tobacco exposure is also more common in countries with high SDI. Thus, regional and national measures, such as some interventions and science education, should be undertaken to increase the availability and consumption of healthy lifestyles rather than tobacco use.

The role of passive smoking in the development of pancreatic cancer remains unclear. The study showed that maternal smoking significantly increases the risk of pancreatic cancer (relative risk = 1.42; 95% CI 1.07 to 1.89) [32]. However, a meta-analysis showed that environmental tobacco smoke was not associated with pancreatic cancer risk during either childhood or adulthood [33]. The effect of secondhand smoke on the risk of pancreatic cancer needs further study. There is also an indirect relationship between smoking and pancreatic cancer: chronic pancreatitis can cause pancreatic cancer, and smoking is an independent risk factor for chronic pancreatitis [34]. The epidemiological investigation of chronic pancreatitis attributable to tobacco also needs to be further conducted in the future to gather full evidence of the relationship between pancreatic cancer and pancreatitis.

The limitations of this study are as follows. First, there is a lack of high-quality and detailed data in some regions and countries/territories, especially in low-income areas, leading to bias in some information in the registry database. Second, the lack of an effective early diagnostic method is another limitation in low-income areas. High-income areas have better health care standards, leading to rich and available records of pancreatic cancer incidence and death. In addition, due to the limitation of the data, we could not conduct in-depth research on tobacco types and exposure doses.

There is plenty of room for improvement in health care interventions and public management policies on tobacco to have a positive influence on pancreatic cancer prevention. Future studies should focus on further expanding global data on the disease burden of pancreatic cancer attributable to tobacco, especially in countries and regions with a high disease burden. It is necessary to establish surveillance systems for the burden of disease attributable to tobacco to enhance the accuracy and reliability of data. Further research on the mechanism by which smoking affects pancreatic cancer is needed to provide more evidence for the early screening and treatment of pancreatic cancer and improve the prognosis of pancreatic cancer. Moreover, the economic burden of tobacco-related pancreatic cancer also deserves attention in future studies.

## 5. Conclusions

Globally, smoking remains the leading behavioral risk factor for pancreatic cancer. The disease burden of smoking-attributable pancreatic cancer is higher in males than in females. The burden of pancreatic cancer attributable to smoking remains higher in high-income regions. Future measures should be taken to limit tobacco use among the population. The early screening and diagnosis of pancreatic cancer in high-risk groups is also needed.

## Figures and Tables

**Figure 1 ijerph-20-01552-f001:**
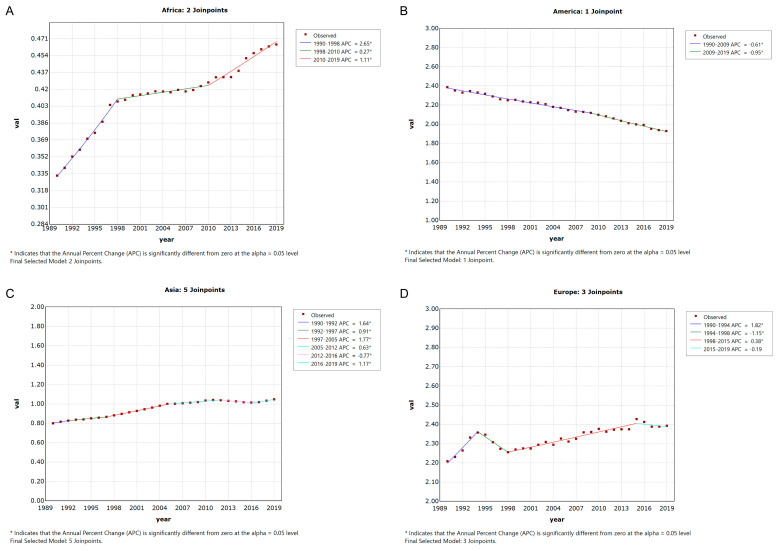
The change trend in the ASDR of pancreatic cancer attributable to smoking in four continents from 1990 to 2019. (**A**) Africa; (**B**) America; (**C**) Asia; (**D**) Europe. APC: annual percentage change.

**Figure 2 ijerph-20-01552-f002:**
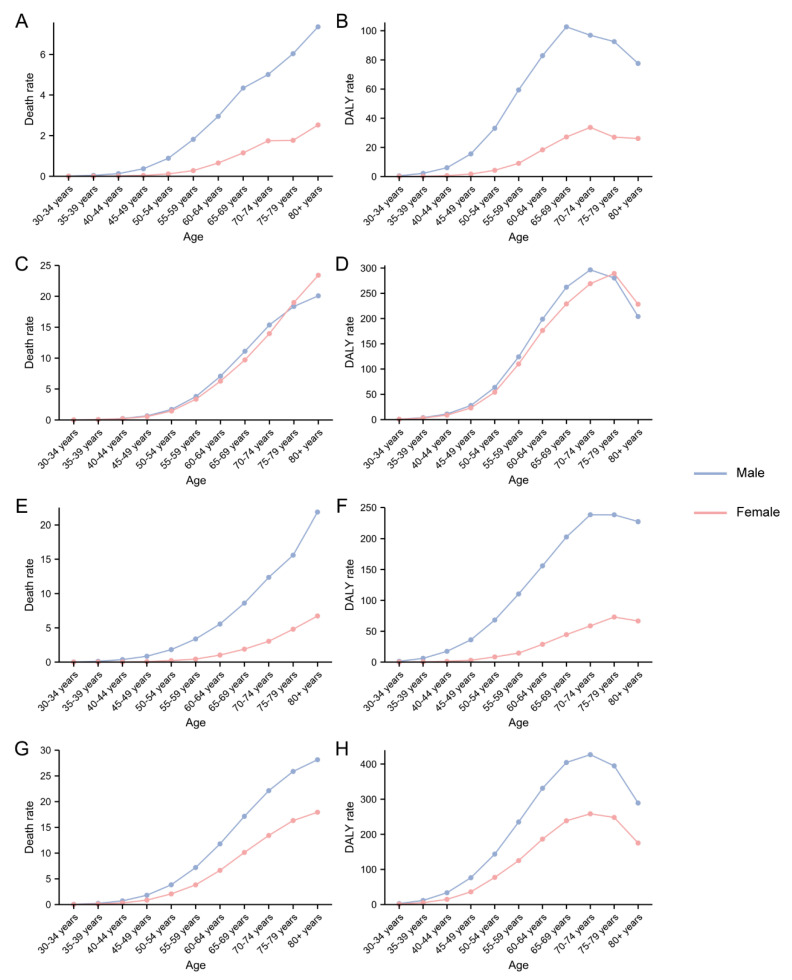
The death and DALY rate of smoking related to pancreatic cancer among four world regions in 2019. (**A**,**B**): Africa; (**C**,**D**): America; (**E**,**F**): Asia; (**G**,**H**): Europe. DALY: disability-adjusted life-year.

**Figure 3 ijerph-20-01552-f003:**
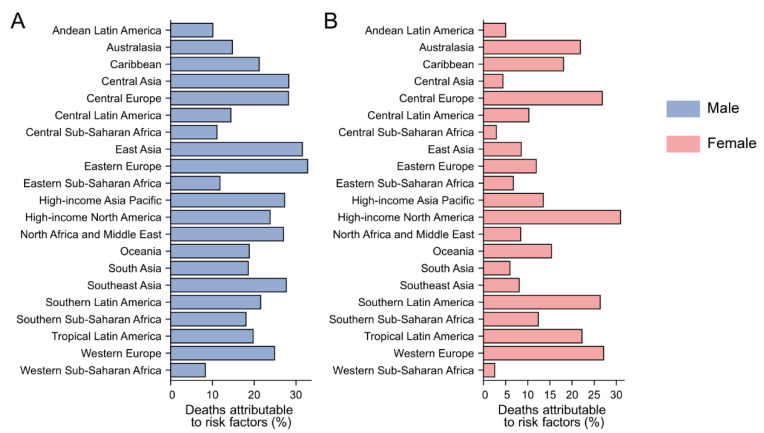
Fraction of pancreatic cancer age-standardized deaths attributable to smoking by region. (**A**) Male; (**B**) Female.

**Figure 4 ijerph-20-01552-f004:**
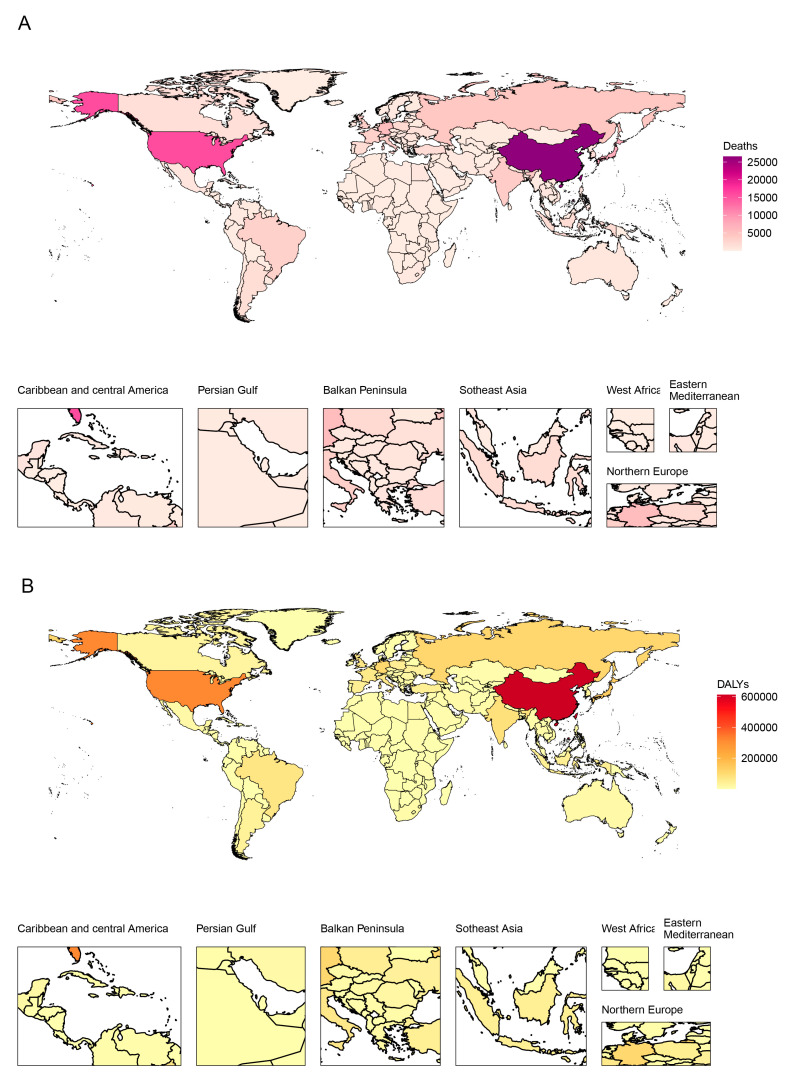
Deaths (**A**) and DALYs (**B**) of pancreatic cancer attributable to smoking across all countries/territories in 2019. DALYs: disability-adjusted life-years.

**Figure 5 ijerph-20-01552-f005:**
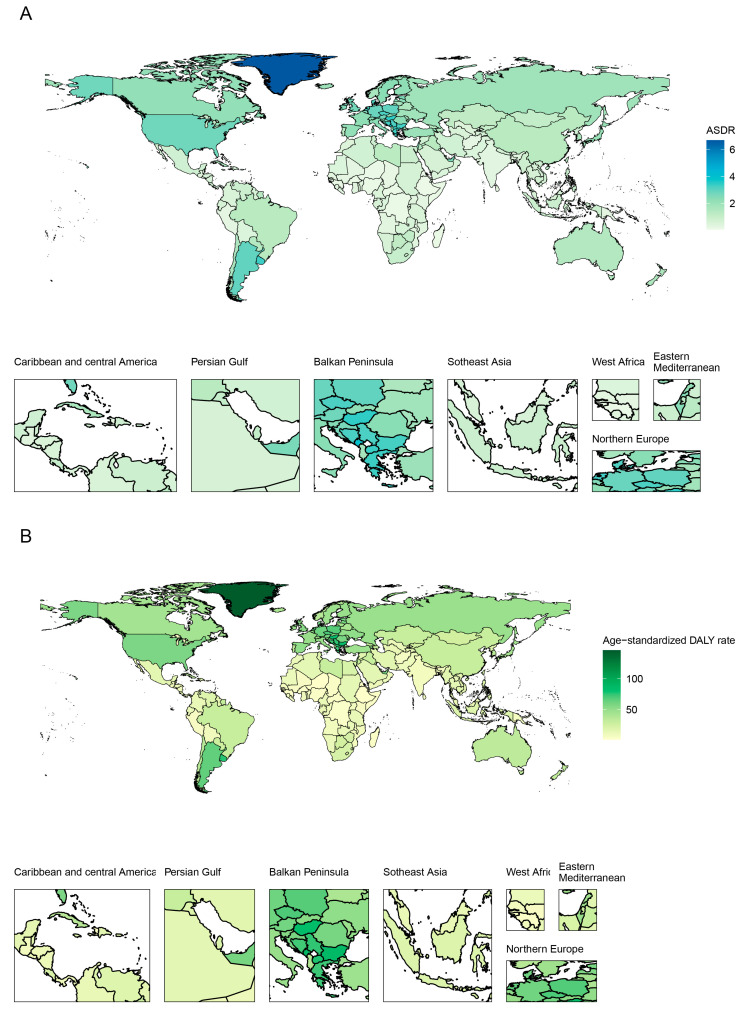
Age-standardized rates of death (**A**) and DALY (**B**) of pancreatic cancer attributable to smoking across all countries/territories in 2019. ASDR: age-standardized death rate; DALY: disability-adjusted life-year.

**Figure 6 ijerph-20-01552-f006:**
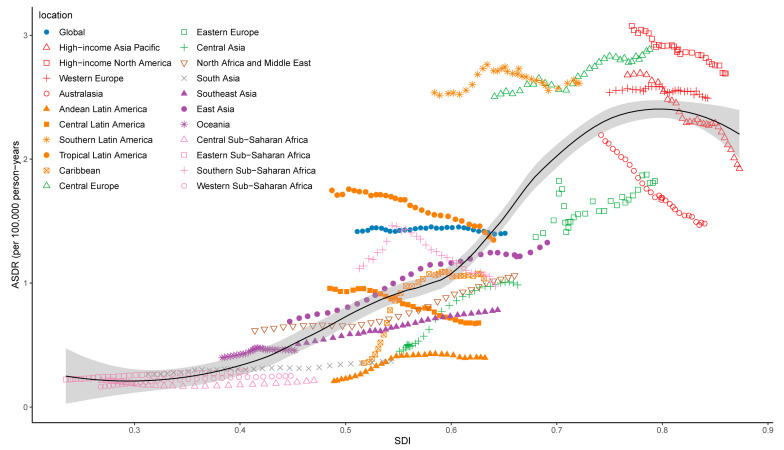
The trend in ASDR of pancreatic cancer attributable to smoking across 21 GBD regions by SDI, from 1990 to 2019. ASDR: age-standardized death rate; SDI: sociodemographic index.

**Figure 7 ijerph-20-01552-f007:**
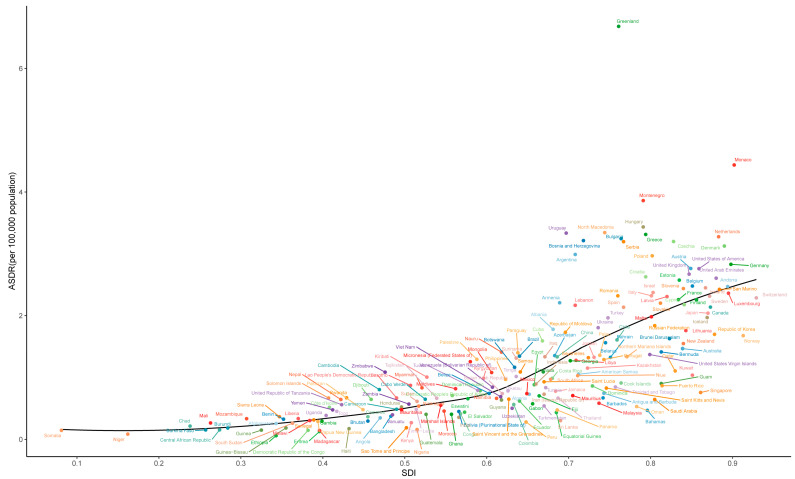
The relationship between ASDR and SDI among all countries/territories in 2019. ASR: age-standardized rate. ASR: age-standardized rate; SDI: sociodemographic index.

**Table 1 ijerph-20-01552-t001:** The age-standardized SEV rates and ARC of smoking among all regions, 1990 to 2019.

	Age-Standardized SEV Rate in 1990	Age-Standardized SEV Rate in 2019	ARC from 1990 to 2019
Global	14.85 (13.27 to 16.56)	11.14 (9.93 to 12.54)	−0.25 (−0.26 to −0.24)
Different SDI			
High SDI	21.17 (19.19 to 23.41)	14.22 (12.71 to 15.94)	−0.33 (−0.35 to −0.31)
High-middle SDI	16.73 (14.95 to 18.68)	14.36 (12.85 to 16.11)	−0.14 (−0.16 to −0.13)
Middle SDI	13.1 (11.53 to 14.76)	10.73 (9.51 to 12.1)	−0.18 (−0.2 to −0.16)
Low-middle SDI	10.76 (9.38 to 12.29)	8.13 (7.07 to 9.34)	−0.24 (−0.26 to −0.22)
Low SDI	6.28 (5.31 to 7.33)	5.03 (4.24 to 5.93)	−0.2 (−0.22 to −0.18)
GBD Region			
Andean Latin America	4.9 (4.04 to 5.88)	3.44 (2.76 to 4.22)	−0.3 (−0.34 to −0.25)
Australasia	19.18 (16.89 to 21.83)	11.92 (10.32 to 13.65)	−0.38 (−0.41 to −0.35)
Caribbean	11.37 (9.96 to 12.88)	8.8 (7.68 to 10.07)	−0.23 (−0.27 to −0.19)
Central Asia	12.41 (11.02 to 13.93)	10.77 (9.54 to 12.1)	−0.13 (−0.17 to −0.1)
Central Europe	21.8 (19.4 to 24.43)	17.7 (15.71 to 19.93)	−0.19 (−0.21 to −0.16)
Central Latin America	10.5 (8.75 to 12.43)	5.33 (4.38 to 6.45)	−0.49 (−0.53 to −0.46)
Central Sub-Saharan Africa	4.45 (3.64 to 5.37)	3.68 (3.03 to 4.42)	−0.17 (−0.23 to −0.11)
East Asia	15.16 (13.46 to 16.95)	14.43 (12.91 to 16.01)	−0.05 (−0.08 to −0.01)
Eastern Europe	16.02 (13.99 to 18.26)	17.38 (15.31 to 19.75)	0.09 (0.04 to −0.13)
Eastern Sub-Saharan Africa	4.8 (3.93 to 5.79)	3.98 (3.24 to 4.81)	−0.17 (−0.2 to −0.14)
High-income Asia Pacific	19.88 (17.79 to 22.21)	13.38 (11.82 to 15.07)	−0.33 (−0.36 to −0.29)
High-income North America	23.3 (21.18 to 25.56)	14.41 (12.99 to 15.97)	−0.38 (−0.41 to −0.36)
North Africa and Middle East	13.37 (12.08 to 14.78)	11.61 (10.44 to 12.92)	−0.13 (−0.16 to −0.11)
Oceania	11.95 (9.61 to 14.45)	9.71 (7.69 to 11.98)	−0.19 (−0.28 to −0.09)
South Asia	9.12 (7.79 to 10.55)	6.32 (5.39 to 7.38)	−0.31 (−0.34 to −0.28)
Southeast Asia	12.87 (11.13 to 14.86)	11.04 (9.45 to 12.87)	−0.14 (−0.17 to −0.11)
Southern Latin America	18.58 (16.01 to 21.39)	14.49 (12.42 to 16.78)	−0.22 (−0.26 to −0.18)
Southern Sub-Saharan Africa	13.55 (11.67 to 15.71)	7.33 (6.21 to 8.63)	−0.46 (−0.49 to −0.42)
Tropical Latin America	18.95 (16.62 to 21.55)	9.27 (8.05 to 10.76)	−0.51 (−0.54 to −0.48)
Western Europe	21.66 (19.47 to 24.18)	16.11 (14.19 to 18.19)	−0.26 (−0.28 to −0.23)
Western Sub-Saharan Africa	3.74 (3.09 to 4.47)	3.15 (2.62 to 3.75)	−0.16 (−0.19 to −0.12)

SEV: summary exposure value; ARC: annualized rates of change; SDI: sociodemographic index.

**Table 2 ijerph-20-01552-t002:** Deaths and ASDR of smoking-attributable pancreatic cancer among all regions, 1990 and 2019.

	Deaths in 1990	ASDR in 1990	Deaths in 2019	ASDR in 2019
Global				
Both sex	53,204 (47,828 to 58,737)	1.42 (1.26 to 1.57)	113,384 (98,830 to 128,466)	1.40 (1.22 to 1.59)
Male	32,166 (28,168 to 36,488)	1.91 (1.66 to 2.17)	72,682 (60,495 to 85,113)	1.95 (1.62 to 2.29)
Female	21,038 (17,427 to 24,318)	1.02 (0.84 to 1.18)	40,702 (32,751 to 48,753)	0.93 (0.75 to 1.11)
Different SDI				
High SDI	28,901 (25,403 to 32,273)	2.72 (2.4 to 3.03)	45,959 (39,751 to 52,864)	2.37 (2.06 to 2.71)
High–middle SDI	16,276 (14,422 to 18,107)	1.53 (1.36 to 1.7)	37,078 (32,078 to 42,315)	1.8 (1.56 to 2.06)
Middle SDI	5595 (4706 to 6530)	0.58 (0.5 to 0.68)	21,892 (18,083 to 26,237)	0.91 (0.76 to 1.09)
Low–middle SDI	1971 (1511 to 2482)	0.37 (0.28 to 0.46)	7062 (5851 to 8267)	0.55 (0.46 to 0.64)
Low SDI	442 (302 to 600)	0.21 (0.15 to 0.28)	1343 (1026 to 1667)	0.29 (0.23 to 0.36)
GBD Region				
Andean Latin America	40 (31 to 52)	0.21 (0.16 to 0.27)	217 (158 to 292)	0.40 (0.29 to 0.53)
Australasia	523 (457 to 589)	2.2 (1.92 to 2.47)	739 (621 to 877)	1.48 (1.25 to 1.74)
Caribbean	91 (78 to 104)	0.36 (0.31 to 0.41)	537 (430 to 659)	1.04 (0.83 to 1.27)
Central Asia	210 (177 to 247)	0.45 (0.38 to 0.53)	706 (598 to 826)	0.99 (0.84 to 1.15)
Central Europe	3753 (3329 to 4163)	2.51 (2.22 to 2.77)	6233 (5281 to 7340)	2.89 (2.46 to 3.4)
Central Latin America	743 (616 to 862)	0.96 (0.8 to 1.1)	1556 (1199 to 1988)	0.68 (0.52 to 0.86)
Central Sub-Saharan Africa	46 (28 to 68)	0.21 (0.13 to 0.31)	106 (68 to 150)	0.21 (0.14 to 0.29)
East Asia	5759 (4601 to 7098)	0.69 (0.56 to 0.83)	27,348 (21,521 to 33,938)	1.33 (1.05 to 1.64)
Eastern Europe	3936 (3371 to 4535)	1.37 (1.17 to 1.58)	6287 (5300 to 7409)	1.82 (1.54 to 2.15)
Eastern Sub-Saharan Africa	149 (101 to 203)	0.22 (0.15 to 0.3)	411 (286 to 549)	0.29 (0.21 to 0.38)
High-income Asia Pacific	5365 (4759 to 6026)	2.68 (2.37 to 3.02)	9007 (7614 to 10,401)	1.92 (1.65 to 2.2)
High-income North America	11,018 (9364 to 12,740)	3.08 (2.62 to 3.54)	17,315 (14,379 to 20,528)	2.69 (2.26 to 3.17)
North Africa and Middle East	1017 (811 to 1276)	0.62 (0.49 to 0.77)	4338 (3598 to 5209)	1.06 (0.88 to 1.28)
Oceania	11 (8 to 14)	0.4 (0.3 to 0.52)	29 (21 to 39)	0.46 (0.34 to 0.6)
South Asia	1290 (886 to 1731)	0.27 (0.19 to 0.35)	4788 (3639 to 5927)	0.37 (0.29 to 0.45)
Southeast Asia	1197 (984 to 1404)	0.51 (0.42 to 0.59)	4420 (3310 to 5837)	0.78 (0.59 to 1.03)
Southern Latin America	1178 (999 to 1343)	2.54 (2.15 to 2.9)	2199 (1867 to 2541)	2.61 (2.22 to 3.02)
Southern Sub-Saharan Africa	289 (226 to 377)	1.12 (0.88 to 1.46)	515 (414 to 619)	0.98 (0.79 to 1.16)
Tropical Latin America	1482 (1298 to 1667)	1.75 (1.52 to 1.98)	3211 (2710 to 3744)	1.35 (1.13 to 1.57)
Western Europe	14,973 (13,254 to 16,698)	2.54 (2.25 to 2.82)	22,993 (19,973 to 26,393)	2.49 (2.18 to 2.84)
Western Sub-Saharan Africa	135 (94 to 185)	0.16 (0.12 to 0.22)	429 (302 to 575)	0.25 (0.18 to 0.33)

ASDR: age-standardized death rate; SDI: sociodemographic index.

**Table 3 ijerph-20-01552-t003:** The AAPC of ASDR and age-standardized DALY rate of smoking-attributable pancreatic cancer among all regions, 1990 to 2019.

Location	AAPC and 95% CI (Death)	*p* Value	AAPC and 95% CI (DALY)	*p* Value
Global	−0.028 (−0.136 to 0.079)	0.606	−0.131 (−0.226 to −0.035)	0.007
SDI				
High SDI	−0.486 (−0.586 to −0.386)	*p* < 0.001	−0.606 (−0.708 to −0.504)	*p* < 0.001
High–middle SDI	0.538 (0.401 to 0.676)	*p* < 0.001	0.368 (0.231 to 0.505)	*p* < 0.001
Middle SDI	1.532 (1.456 to 1.607)	*p* < 0.001	1.386 (1.316 to 1.456)	*p* < 0.001
Low–middle SDI	1.424 (1.208 to 1.639)	*p* < 0.001	1.362 (1.308 to 1.417)	*p* < 0.001
Low SDI	1.176 (1.086 to 1.266)	*p* < 0.001	1.071 (0.956 to 1.185)	*p* < 0.001
GBD regions				
Andean Latin America	2.288 (2.022 to 2.554)	*p* < 0.001	2.088 (1.787 to 2.39)	*p* < 0.001
Australasia	−1.331 (−1.496 to −1.166)	*p* < 0.001	−1.377 (−1.517 to −1.236)	*p* < 0.001
Caribbean	3.849 (3.31 to 4.391)	*p* < 0.001	3.785 (3.224 to 4.349)	*p* < 0.001
Central Asia	2.822 (2.39 to 3.257)	*p* < 0.001	2.558 (2.162 to 2.955)	*p* < 0.001
Central Europe	0.492 (0.33 to 0.655)	*p* < 0.001	0.354 (0.183 to 0.526)	*p* < 0.001
Central Latin America	−1.193 (−1.482 to −0.903)	*p* < 0.001	−1.246 (−1.505 to −0.986)	*p* < 0.001
Central Sub-Saharan Africa	−0.019 (−0.22 to 0.182)	0.853	0.004 (−0.205 to 0.214)	0.969
East Asia	2.253 (2.009 to 2.497)	*p* < 0.001	2.052 (1.842 to 2.262)	*p* < 0.001
Eastern Europe	1.195 (0.569 to 1.824)	*p* < 0.001	0.949 (0.046 to 1.861)	0.039
Eastern Sub-Saharan Africa	0.914 (0.779 to 1.049)	*p* < 0.001	0.872 (0.739 to 1.005)	*p* < 0.001
High-income Asia Pacific	−1.149 (−1.27 to −1.027)	*p* < 0.001	−1.267 (−1.377 to −1.156)	*p* < 0.001
High-income North America	−0.452 (−0.609 to −0.294)	*p* < 0.001	−0.641 (−0.837 to −0.446)	*p* < 0.001
North Africa and Middle East	1.874 (1.711 to 2.037)	*p* < 0.001	1.72 (1.53 to 1.91)	*p* < 0.001
Oceania	0.497 (0.38 to 0.615)	*p* < 0.001	0.451 (0.389 to 0.512)	*p* < 0.001
South Asia	1.211 (0.601 to 1.825)	*p* < 0.001	1.155 (0.961 to 1.35)	*p* < 0.001
Southeast Asia	1.522 (1.423 to 1.62)	*p* < 0.001	1.317 (1.198 to 1.435)	*p* < 0.001
Southern Latin America	0.146 (−0.075 to 0.367)	0.195	0.005 (−0.174 to 0.184)	0.954
Southern Sub-Saharan Africa	−0.482 (−0.921 to −0.042)	0.032	−0.617 (−1.134 to −0.097)	0.02
Tropical Latin America	−0.89 (−1.044 to −0.736)	*p* < 0.001	−0.955 (−1.111 to −0.799)	*p* < 0.001
Western Europe	−0.068 (−0.206 to 0.07)	0.333	−0.156 (−0.297 to −0.014)	0.031
Western Sub-Saharan Africa	1.517 (1.437 to 1.597)	*p* < 0.001	1.444 (1.314 to 1.574)	*p* < 0.001

AAPC: average annual percentage change; DALY: disability-adjusted life-year; SDI: sociodemographic index.

## Data Availability

The data in this study can be found in the Global Health Data Exchange query tool (https://vizhub.healthdata.org/gbd-results/ (accessed on 31 October 2022)).

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
