# Peer review of "The Global, Regional and National Burden of Pancreatic Cancer Attributable to Smoking, 1990 to 2019: A Systematic Analysis from the Global Burden of Disease Study 2019"

_ijerph, 2023, doi:10.3390/ijerph20021552_

Round 1

Reviewer 1 Report

Thank you for your submission of your systematic analysis of the global, regional, and national burden of pancreatic cancer and the relationship with tobacco use. The analysis is very informative and interesting.

Reviewer 2 Report

This is a well designed study, with interesting conclusions. Study design and methods are great. However, i think that is a terrible omission to forget to say somehting about laryngeal cancer. SCC of larynx is strongly related to smoking. Authors should include in the 39-40 line information about squamous cell carcinoma of larynx and head and neck cancer. These articles provide all the necessary information and can be added.

1) Tsetsos N, Poutoglidis A, Vlachtsis K, Stavrakas M, Nikolaou A, Fyrmpas G. Twenty-year experience with salvage total laryngectomy: lessons learned. J Laryngol Otol. 2021 Aug;135(8):729-736

2) Tsentemeidou A, Fyrmpas G, Stavrakas M, Vlachtsis K, Sotiriou E, Poutoglidis A, Tsetsos N. Human Papillomavirus Vaccine to End Oropharyngeal Cancer. A Systematic Review and Meta-Analysis. Sex Transm Dis. 2021 Sep 1;48(9):700-707.

3) Obid R, Redlich M, Tomeh C. The Treatment of Laryngeal Cancer. Oral Maxillofac Surg Clin North Am. 2019 Feb;31(1):1-11

4) Steuer CE, El-Deiry M, Parks JR, Higgins KA, Saba NF. An update on larynx cancer. CA Cancer J Clin. 2017 Jan;67(1):31-50

After these minor modification, I storngly recommend the publication of this study.

Reviewer 3 Report

The paper entitled “The global, Regional and National Burden of Pancreatic Cancer Attributable to Smoking, 1990 to 2019: A Systematic Analysis from the Global Burden of Disease Study 2019” is an interesting collection. However, the relation between smoking and pancreatic cancer incidence must be proved by meta-analysis. The greatest weakness of this paper is the simple data collection and the need thorough statistical analysis which must follow the the PRISMA guidelines.

Round 2

Reviewer 3 Report

Can be accepted in the present Form